# Anti-Leukemic Effects Induced by Dendritic Cells of Leukemic Origin from Leukemic Blood Samples Are Comparable under Hypoxic vs. Normoxic Conditions

**DOI:** 10.3390/cancers16132383

**Published:** 2024-06-28

**Authors:** Fatemeh Doraneh-Gard, Daniel Christoph Amberger, Carina Amend, Melanie Weinmann, Christoph Schwepcke, Lara Klauer, Olga Schutti, Hedayatollah Hosseini, Doris Krämer, Andreas Rank, Christoph Schmid, Helga Maria Schmetzer

**Affiliations:** 1Medical Department III, Working-group Immune-Modulation, Klinikum Großhadern, Ludwig-Maximilians-University, 81377 Munich, Germany; nikoo_792000@yahoo.com (F.D.-G.); carina_gunsilius@web.de (C.A.); melli.weinmann@hotmail.de (M.W.); christophkugler@gmx.net (C.S.); lara.klauer@kabelmail.de (L.K.); olga.schutti@gmail.com (O.S.); 2Bavarian Cancer Research Center (BZKF), 86156 Augsburg, Germany; andreas.rank@uk-augsburg.de (A.R.); christoph.schmid@uk-augsburg.de (C.S.); 3First Department of Medicine, Paracelsus Medical University, 5020 Salzburg, Austria; 4Experimental Medicine and Therapy Research Department, Faculty of Medicine, University of Regensburg, 93040 Regensburg, Germany; hedayatollah.hosseini@klinik.uni-regensburg.de; 5Department of Hematology, Oncology and Palliative Care, Hospital Hagen, 58097 Hagen, Germany; kraemerd@kkh-hagen.de; 6Department of Hematology and Oncology, University Hospital of Augsburg, 86156 Augsburg, Germany

**Keywords:** DC_leu_, AML, hypoxia

## Abstract

**Simple Summary:**

Dendritic cells are important mediators in the activation of the innate and adoptive immune response. The generation of DC/DC_leu_ was comparable under physiological hypoxic and normoxic culture conditions, with no significant differences in frequencies of generated DC/DC_leu_ from leukemic cell lines, peripheral blood mononuclear as well as whole blood cells from patients with acute myeloid leukemia. Moreover, the composition of immune cells and anti-leukemic immune activity after T cell enriched mixed lymphocyte culture with patients’ blood pretreated vs not pretreated with defined response modifiers (Kits) was improved, particularly under hypoxic conditions. These data show that the generation of DC/DC_leu_ as well as the induction of anti-leukemic immune activating functionality is possible under standard normoxic as well as physiological hypoxic culture conditions ex vivo.

**Abstract:**

Hypoxia can modulate the immune system by affecting the function and activity of immune cells, potentially leading to altered immune responses. This study investigated the generation of leukemia-derived dendritic cells (DC_leu)_ from leukemic blasts and their impact on immune cell activation under hypoxic (5–10% O_2_) compared to normoxic (21% O_2_) conditions using various immunomodulatory Kits. The results revealed that DC/DC_leu_-generation was similar under hypoxic and normoxic conditions, with no significant differences observed in frequencies of generated DC/DC_leu_. Furthermore, the study showed that the activation of immune cells and their anti-leukemic activity improved when T cell-enriched immunoreactive cells were co-cultured with DC/DC_leu_ which were generated with Kit I and M compared to the control after mixed lymphocyte cultures. The anti-leukemic activity was improved under hypoxic compared to normoxic conditions after MLC^WB-DC Kit M^. These findings suggest that DC/DC_leu_-cultures of leukemic whole blood with Kits under hypoxic conditions yield comparable frequencies of DC/DC_leu_ and can even increase the anti-leukemic activity compared to normoxic conditions. Overall, this research highlights the potential of utilizing DC/DC_leu_ (potentially induced in vivo with Kits) as a promising approach to enhance immune response in patients with acute myeloid leukemia.

## 1. Introduction

Acute myeloid leukemia (AML) is a clonal hematological disorder with an overall five-year survival rate in elderly patients of about 28% [1]. Although 60–80% of all patients achieve a complete remission (CR), 70–80% of these patients relapse in the following 2 years [2]. Ongoing clinical trials analyze new immunotherapeutic strategies and immunomodulatory approaches to target leukemic cells, to stabilize remissions and to improve the outcome of patients with AML [3,4].

Dendritic cells (DCs) play a pivotal role as specialized antigen presenting cells (APCs) on the interface of the innate and adoptive immune system [5,6,7]. The generation of DC/DC_leu_ ex vivo from blast-containing peripheral blood mononuclear cells (PBMNCs) or directly from leukemic whole blood (WB) represents an immunotherapy option for patients with AML. With different protocols, myeloid progenitor cells can be converted to DC/DC_leu_ presenting DCs’ together with patients’ blast antigens in a costimulatory manner without the induction of blasts’ proliferation [8,9,10,11,12]. DC/DC_leu_-generating protocols or Kits contain different response modifiers such as myelopoiesis stimulating factors [e.g., granulocyte-macrophage-colony stimulating factors (GM-CSF)], danger signaling factors [(e.g., bacterial or nucleic stimuli such as Picibanil (OK432)—a lysis product from the streptococcus pyogenes)] and mediators for the maturation of DC/DC_leu_ [(e.g., Prostaglandin (PGE) E_1_ or E_2_, Tumor-necrosis-factor alpha (TNF-α) or Interleukin 4 (IL 4)] [6,9,10]. In T cell enriched mixed lymphocyte culture (MLC) generated DC/DC_leu_ are known to increase frequencies of non-naïve T cells, proliferating CD71^+^ or CD69^+^ T cells, central memory T cells (T_cm_, CD45RO^+^CCR7^+^), effector (memory) T cells (T_eff-em_, CD45RO^+^CCR7^-^) and to reduce frequencies of naive T cells and regulatory T cells (T_reg_, CD25^++^CD127^low^) [13,14,15,16]. DC/DC_leu_ moreover contribute by activating cells of the innate immune system and cells on the interface of the innate and the adaptive immune system, such as natural killer cells (NK cells, CD56^+^CD3^-^), invariant natural killer cells (iNKT cells, 6B11^+^CD3^+^) or cytokine induced killer cells (CIK cells, CD56^+^CD3^+^) [11,12,17,18,19].

Hematopoietic stem cells, leukemic cells as well as immune cells reside in the bone marrow (BM) under hypoxic O_2_ concentrations of about 0.1–1%, in the arterial blood of about 12% or of 4–15% in peripheral blood (PB) [20,21]. That means that hematopoietic cells are exposed to changing O_2_ concentrations/saturation during their differentiation [22]. It was shown that hypoxic conditions might change the anti-leukemic activity of cells against leukemic blasts [23]. Moreover, reactions of immune cells prepared ex vivo for adoptive transfer under normoxic conditions might not reflect the in vivo situation [22] and might be switched to cell preparations under hypoxic conditions.

The aim of this study was to analyze the impact of hypoxic conditions on the generation of DC/DC_leu_ from leukemic and healthy PBMNCs and WB and furthermore, to analyze the effect of DC/DC_leu_ stimulation on the anti-leukemic immune activation after MLC in comparison to normoxic conditions.

## 2. Material and Methods

### 2.1. Sample Collection

After obtaining written informed consent in accordance with the local Ethics Committee (Pettenkoferstraße 8a, 80336 Munich, Ludwig Maximilians University Hospital in Munich; Vote No 339-05), patients’ PB or BM samples were provided by the University Hospital of Oldenburg, Tübingen and Augsburg. Anticoagulation was performed with Lithium heparintubes (7.5 mL, Sarstedt, Nuernberg, Germany) containing standardized concentrations of Heparin. PBMNCs were separated by density gradient centrifugation (density gradient 1.077 g/mL) using the Ficoll Hypaque technique. CD3^+^T cells were positively selected by using the MACS technology (Milteney Biotech, Bergisch Gladbach, Germany) according to manufacturers’ instructions. The purity of the resulting T cells was on average 90% (range 81–97%). Cells were frozen at −80 °C (using DMSO) and thawed according to standardized protocols.

### 2.2. Patients’ Characteristics and Diagnostics

DC/DC_leu_ were generated from AML cell lines, PBMNCs- and WB-samples obtained from AML patients (*n* = 34) in acute phases of the disease and from healthy donors (*n* = 16). Average age of AML patients was 59 years (range 21–78) and of healthy volunteers 28 years (range 21–56). The female to male ratio of AML patients was 1:1.2 and of healthy volunteers 1:1.3.

The following cell lines were included in analysis: Mono-Mac 6 (AML-M5), THP-1 (AML-M5), KG-1 (AML-M4) and NB-4 (M3). These cell lines were purchased from the DSMZ (German collection of Microorganisms Cell Cultures, Braunschweig, Germany) and were cultured according to the manufacturers’ instructions (Table 1).

The classification and diagnosis of patients was based on the FAB (French–American–British) classification: AML without maturation (M1: *n* = 6), AML with granulocytic maturation (M2: *n* = 7), acute myelomonocytic leukemia (M4: *n* = 8) and acute monocytic leukemia (M5: *n* = 8). No FAB classification was available in 5 AML cases (*n* = 5). Patients presented with primary AML [pAML (*n* = 24)] or with secondary AML [sAML (*n* = 8)]. In two cases, no information was available. Patients’ stages were: first diagnosis (*n* = 29), relapse (*n* = 4) or relapse after HSCT (*n* = 1). Patients’ characteristics are given in Table 1.

### 2.3. Cellular Composition of Uncultured Blood Samples Used for Subsequent Experiments

AML WB samples contained on average 32.0% (e.g., CD34^+^, CD65^+^ and/or CD117^+^) leukemic blasts (range 7.2–92.0%), 13.8% CD3^+^T cells (range 3.5–18.9%), 2.3% CD19^+^B cells (range 0.4–8.4%), 5.8% CD56^+^CD3^-^NK cells (range 0.8–7.6%) and 3.1% CD14^+^monocytes (range 0.5–5.1%).

AML PBMNCs samples contained on average 51.3% leukemic blasts (range 21.0–62.0%), 4.2% CD3^+^T cells (range 0.3–17.3%), 0.9% CD19^+^B cells (range 0.2–1.9%), 1.4% CD56^+^CD3^-^NK-cells (range 0.1–3.0%) and 2.3% CD14^+^monocytes (range 0.2–4.2%).

Quantification of NK cells and monocytes was not possible in cases with aberrant expression of CD14 or CD 56 on leukemic blasts.

Healthy WB samples contained on average 5.6% CD14^+^monocytes (range 4.4–8.5%), 19.4% CD3^+^T cells (range 13.6–26.3%), 4.3% CD56^+^CD3^-^NK-cells (range 2.3–6.6%) and 2.3% CD19^+^B cells (range 0.8–4.8%).

Healthy PBMNCs samples contained on average 6.6% CD14^+^monocytes (range 2.6–9.4%), 33.3% CD3^+^T cells (range 27.1–46.7%), 4.8% CD56^+^CD3^-^NK-cells (range 0.1–4.3%) and 2.1 CD19^+^B cells (range 1.3–3.3%).

### 2.4. Cell Cultures Experiments

Human DC/DC_leu_-cultures, MLC-cultures as well as the cytotoxicity fluorolysis assay were set up in parallel under standard normoxic laboratory conditions (37 °C, 21% O_2_ and 5% CO_2_) as well as under hypoxic conditions (37 °C, 10% O_2_ and 5% CO_2_) [22,24]. Cell-culture experiments were set up under 37 °C, 5% O_2_ and 5% CO_2_. For the hypoxic experiments, the hypoxic Workstation INVIVO2 400 from Ruskinn Technology (Bridgend, United Kingdom) was used.

### 2.5. Generation of DC/DC_leu_ from AML Cell Lines

DC/DC_leu_ were generated from AML cell lines with the DC/DC_leu_-generating protocols Kit I and Kit M [6,25]. Therefore, 3–4 × 10^6^ cells were pipetted in 12-multiwell plates in 2 ml serum-free X-Vivo15-medium (Lonza, Basel, Swiss) and were plated into 12-multiwell plates (Thermo Fisher Scientific, Darmstadt, Germany). Response modifiers and immune modulating factors were added to cultures as described below. A culture without added response modifiers served as a control. All response modifiers used for the DC/DC_leu_ generation are approved for human treatment. Compositions of DC/DC_leu_-generating protocols are given in Table 2.

### 2.6. Generation of DC/DC_leu_ from Isolated PBMNC or WB

DC/DC_leu_ were generated from 3–4 × 10^6^ PBMNC (isolated by density gradient centrifugation) from healthy donors and AML patients’ blood using Pici-_PGE1_ and Pici-_PGE2_ (Table 2) as described before [6,9,25,26]. PBMNCs were diluted in 2 ml serum-free X-Vivo15-medium (Lonza, Basel, Swiss) and were plated into 12-multiwell plates (Thermo Fisher Scientific, Darmstadt, Germany). Response modifiers were added as described below. Half medium exchange was carried out after 3–4 cell culture days. A culture without added response modifiers served as a control.

DC/DC_leu_ were generated from healthy and leukemic WB (presenting the physiological cellular and soluble composition of the individual samples) with the DC/DC_leu_-generating protocols Kit I and Kit M, as described before [6,25]. A culture without added response modifiers served as a control. All response modifiers used in Kits for the DC/DC_leu_ generation are approved for human treatment. Compositions of DC/DC_leu_-generating protocols are given in Table 2.

#### 2.6.1. Picibanil-PGE_1_ (Pici_-PGE1_)

DC/DC_leu_ were generated from PBMNCs with the DC/DC_leu_-generating protocol Pici-PGE_1,_ containing 500 U/mL GM-CSF (Sanofi-Aventis, Frankfurt, Germany) and 250 U/mL IL 4 (PeproTech, Berlin, Germany). After 6–7 days, 10 μg/mL Picibanil (OK 432)—a lysis product from Streptococcus pyogenes (Chugai Pharmaceutical Co., Kajiwara, Japan) and 1 μg/mL PGE_1_ (PeproTech, Berlin, Germany) were added. After 7–10 days of incubation, cells were harvested and used for subsequent experiments [6,9,27].

#### 2.6.2. Picibanil-PGE_2_ (Pici_-PGE2_)

DC/DC_leu_ were generated from PBMNCs with the Pici-PGE_2_ DC/DC_leu_-generating protocol, with the same composition as given above for Pici_-PGE1_; however, PGE_1_ was substituted by PGE_2_ (PeproTech, Berlin, Germany) [9,27].

#### 2.6.3. Kit I

DC/DC_leu_ were generated with Kit I from WB and cell lines using 800 U/mL GM-CSF and 10 µg/mL Picibanil [25]. After 2–3 days, the same amounts of response modifiers were added and after in total 7–10 days of incubation, cells were harvested and used for subsequent experiments.

#### 2.6.4. Kit M

DC/DC_leu_ were generated with Kit M from WB and cell lines using 800 U/mL GM-CSF and 1 μg/mL PGE_1_ [25]. Incubations were performed in analogy to Kit I.

**Table 2 cancers-16-02383-t002:** DC/DC_leu_-generating protocols.

DC/DC_leu_-Generating Protocol	Component	Concentration	Sources of DC/DC_leu_	Mode of Action	Culture Time	References
Pici-_PGE1_	GM-CSF	500 U/mL	PBMNC	GM-CSF: induction of myeloid (DC-) differentiation	7–10 Days	[6]
	IL 4	250 U/mL				
	OK-432	10 μg/mL				
	PGE_1_	1 μg/mL				
Pici-_PGE2_	GM-CSF	500 U/mL	PBMNC	IL-4: induction of DC-differentiation/maturation	7–10 Days	[6,28]
	IL 4	250 U/mL				
	OK-432	10 μg/mL				
	PGE_2_	1 μg/mL				
Kit I *	GM-CSF	800 U/mL	WB/ Cell lines	Picibanil (OK-432): lysis product from streptococcus pyogenes; stimulates DC-differentiation	7–10 Days	[26]
	OK-432	1 μg/mL				
Kit M *	GM-CSF	800 U/mL	WB/ Cell lines	PGE1: increases CCR7-expression and enhances DC/DC_leu_-maturation	7–10 Days	[26]
	PGE_1_	1 μg/mL				

DC dendritic cells; DC_leu_ dendritic cells of leukemic origin; GM-CSF granulocyte macrophage colony stimulating factor; IL 4 interleukin 4; OK-432 Picibanil; PGE_2_ prostaglandin E2; PGE_1_ prostaglandin E1; PBMNCs peripheral blood mononuclear cells; WB whole blood; * European Patent 15 801 987.7-1118 and US Patent 15-517627.

### 2.7. Cell-Characterization by Flow Cytometry

Leukemic blasts, T cell subsets, B cells, monocytes and DC/DC_leu_ subsets in the PBMNC- and WB-fractions were quantified using flow cytometry, as described previously. Panels with several monoclonal antibodies (moAbs) labeled with Fluorescein isothiocyanate (FITC), phycoerythrin (PE), tandem Cy7-PE conjugation (Cy7-PE), or allophycocyanin (APC) were used. Antibodies were provided by Beckman Coulter, Krefeld, Germany (^a^), Becton Dickinson, Heidelberg, Germany (^b^), Miltenyi Biotech, Bergisch Gladbach, Germany (^c^), Thermo Fisher, Darmstadt, Germany (^d^) and Santa Cruz Biotechnology, Heidelberg, Germany (^e^). FITC-conjugated moAbs against CD3^a^, CD15^a^, CD33^a^, CD34^a^, CD45RO^a^, CD65^a^, CD71^a^, CD83^a^, CD161^b^ and IPO-38^e^ were used. To detect CD3^a^, CD4^a^, CD19^a^, CD33^a^, CD34^a^, CD56^a^, CD80^b^, CD83^a^, CD117^a^_,_ CD206^a^ and 6B11^b^, PE-conjugated moAbs were used. MoAbs against CD3^a^, CD4^a^, CD14^b^, CD15^b^, CD19^a^, CD33^a^, CD34^a^, CD56^a^, CD65^c^, CD80^b^, CD117^a^ and CD197^b^ were labeled with Cy7-PE. APC-labeled moAbs against CD3^a^, CD4^b^, CD14^a^, CD15^b^, CD34^a^, CD45RO^d^, CD56^a^, CD65^c^, CD69^b^, CD83^b^, CD86^g^, CD117a, CD206^b^ and CD209^b^ were used. 7AAD^b^ was used to detect dead cells. To stain intracellular antigens (e.g., IPO-38), the FIX & PERM^®^ Cell Fixation and Cell Permeabilization Kit (Thermo Fisher Scientific, Darmstadt, Germany) were used according to manufacturer’s instructions.

For analysis and quantification of DCs and DC_leu_ in the total- or in the cell-subtype fractions after DC/DC_leu_-cultures, we used a refined gating strategy, as shown before [8,9,25]. DC_leu_ are characterized by the simultainous expression of DCs’ together with patients’ blast antigens. To quantify generated DC_leu_, the cells were stained with patients or leukemic cell line specific blast-staining antibodies (e.g., CD34, CD65, and CD117), according to diagnostic reports in combination with DC-staining antibodies (e.g., CD80, CD83, CD86, CD206, and CD209), which were not expressed on blasts before culture. DC_leu_ were quantified in the total cell fraction (DC_leu_/PBMNC or WB), in the DC-fraction (DC_leu_/DC^+^) or in the blast fraction to quantify the amounts of blasts converted to DC_leu_ (DC_leu_/bla^+^). Proliferating blasts were characterized by their co-expression of CD71 or IPO-38 without co-expression of DC-markers (Table 3). According to their expression profiles we quantified frequencies of immune-reactive cells in the total cell fraction (e.g., CD3^+^/cells) or in the subpopulations (e.g., in CD3^+^cells) as given in Table 3.

### 2.8. Mixed Lymphocyte Culture (MLC) of T Cell Enriched Immune Reactive Cells with Kit Treated vs Untreated WB from AML-Patients

A total of 1 × 10^6^ (thawn) autologous CD3^+^T cells from AML patients were co-cultured with IL 2 and a stimulator cell suspension containing approximately 2.5 × 10^5^ generated DC/DC_leu_ which were generated with different DC/DC_leu_-generating protocols from leukemic WB. MLC of T cell enriched immunoreactive cells with a stimulator cell suspension without pretreatment with different DC/DC_leu_-generating protocols (MLC^WB^) served as a control, as shown before [28].

Cells were harvested after 6–7 days; subtypes were quantified by flowcytometry and used for cytotoxicity fluorolysis assay [28].

**Table 3 cancers-16-02383-t003:** Monocytes, leukemic blasts, DC/DC_leu_ and T cell substes as evulated by flow cytometry.

	Name of Subgroup	Referred to	Surface Marker (CD)	Abbreviation	References
Monocytes	CD14^+^ monocytes	PBMNC, WB	CD14^+^		[8,9]
Blasts and DC/DC_leu_	Leukemic blasts	cells (PBMNC, WB)	Bla^+^ (CD15, CD33, CD34, CD65, CD117)	Bla^+^/ cells (PBMNC, WB)	[8,9]
	Dendritic cells	cells (PBMNC, WB)	DC^+^ (CD80, CD83, CD86, CD206, CD209)	DC^+^/cells (PBMNC, WB)	[8,9]
	Leukemia derived DC	cells (PBMNC, WB	DC^+^Bla^+^	DC_leu_/cells (PBMNC, WB)	[8,9]
	DC_leu_ in DC fraction	DC^+^	DC^+^Bla^+^	DC_leu_/DC^+^	[8,9]
	DC_leu_ in leukemic blast fraction	Bla^+^	DC^+^Bla^+^	DC_leu_/Bla^+^	[8,9]
	Mature DC in DC fraction	DC^+^	DC^+^CD197^+^	DC_mat_/DC^+^	[8,9]
	Proliferating leukemic blasts	WB	Bla^+^DC^-^ CD71^+^	Bla_prol-CD71_/Bla^+^	[29]
	Proliferating leukemic blasts	WB	Bla^+^DC^-^ IPO-38^+^	Bla_prol-IPO38_/ Bla^+^	[29]
T cell substes	CD3^+^ pan-T cells	WB	CD3^+^	CD3^+^/cells	[14]
	CD4^+^ T cells	CD3^+^	CD3^+^CD4^+^	CD3^+^CD4^+^/CD3^+^	[14]
	CD8^+^T cells	CD3^+^	CD3^+^CD8^+^	CD3^+^CD8^+^/CD3^+^	[14]
	Naive T cells	CD3^+^	CD3^+^CD45RO^-^	T_naive_/CD3^+^	[14]
	Non-naive T cells	CD3^+^	CD3^+^CD45RO^+^	T_non-naive_/CD3^+^	[14]
	Central (memory) T cells	CD3^+^	CD3^+^CCR7^+^CD45RO^+^	T_cm_/CD3^+^	[14]
	Effector (memory) T cells	CD3^+^	CD3^+^CCR7^-^CD45RO^+^	T_eff-em_/CD3^+^	[14]
	Early proliferating T cells	CD3^+^	CD3^+^CD69^+^	T_prol-early_/CD3^+^	[14]
	Late proliferating T cells	CD3^+^	CD3^+^CD71^+^	T_prol-late_/CD3^+^	[14]
	CD4^+^Regulatory T cells	CD3^+^	CD3^+^CD4^+^CD25^++^CD127^low^	CD4^+^T_reg_/CD4^+^	[14,16]]
CIK cells	Cytokine induced killer cells	MLC	CD3^+^CD56^+^ CD3^+^CD161^+^	CD3^+^CD56^+^/MLC CD3^+^CD161^+^/MLC	[14,16,18,19]
NK cells	Natural killer cells	MLC	CD3^-^CD56^+^ CD3^-^CD161^+^	CD3^-^CD56^+^/MLC CD3^-^CD161^+^/MLC	[14,16,18,19]
iNKT cells	Invariant natural killer cells	MLC	6B11^+^ CD3^+^6B11^+^	6B11^+^/MLC CD3^+^6B11^+^/MLC	[14,16,18,19]

PBMNC peripheral blood mononuclear cells; WB whole blood; CD cluster of differentiation; DC dendritic cells; DC_leu_ dendritic cells of leukemic origin; Bla leukemic blasts; MLC mixed lymphocyte culture.

### 2.9. Cytotoxicity Fluorolysis Assay (CTX)

The Cytotoxicity Fluorolysis Assay was conducted to assess the lytic activity of autologous T cell-enriched immunoreactive cells after stimulation with DC/DC_leu_ containing cell fractions after treatment with Kit M, Kit I or control (without added response modifies) after MLC (‘effector cells’) against autologous leukemic blasts (‘target cells’). Therefore, effector and target cells (with a ratio of 1:1) were co-cultured under hypoxic and normoxic conditions and incubated for 3 and 24 h. Target cells were stained with respective anti-bodies before incubation. After harvest, 7AAD and a defined number of Fluorosphere beads (Beckman Coulter) were added. As a control, effector and target cells were cultured separately and mingled shortly before measurements. Flow cytometric analyses were performed after 3 and 24 h of effector and target cells’ co-incubation using a refined gating strategy [30]. The lytic activity against leukemic target blasts (blast lysis) is defined as the difference in frequencies of viable blasts in the effector target cell cultures as compared to controls. Improved blast lysis is defined as the difference in proportions of blast lysis achieved after MLC with DC/DC_leu_ generated with Kits compared to control.

### 2.10. Cell Cycle Experiments

The cell cycle profile of cell line samples was determined by staining of DNA with PI-fluorescent dyes. PI intercalates into the groove of double-stranded DNA producing a highly fluorescent signal. As PI can also bind to double-stranded RNA, the cells must be treated with RNAse for DNA resolution. We associated a fixation (paraformaldehyde) with PI staining. The PI staining of DNA allowed the detection of cells in G0/G1, S phase, and G2/M as described earlier [31].

### 2.11. Quantitative PCR (Real Time PCR)

Total RNA was isolated from 10^6^ cells of each cell line using MagNA Pure LC mRNA HS Kit (Roche, Basel, Switzerland) according to the manufacturer’s instructions. cDNA was synthesized from 1 μg aliquots of total RNA in a 20 μL standard reaction mixture using SuperScript^®^ III First-Strand Synthesis System for RT-PCR (Invitrogen, Camarillo, CA, USA) according to manufacturer’s instructions. Quantitative Real-time polymerase chain reaction (RT-PCR) was performed using the 7900HT Fast Real-Time PCR System (Applie Biosystems, Waltham, MA, USA) with 2 μL of cDNA, Fast SYBR^®^ Green Master Mix (Applied Biosystems, Waltham, MA, USA) [32]. We checked the expression of the fusion genes related to each cell line in Normoxic vs Hypoxic condition. Furthermore, glyceraldehyde-3-phosphate dehydrogenase gene (GAPDH) was used as a reference for the normalization of ΔCT values.

### 2.12. Statistical Methods

Mean ± standard derivations are given. Statistical comparisons of two groups were performed using the two-tailed t-test (in cases with data normally distributed) and the Mann–Whitney–Wilcoxon Test (in cases with data not normally distributed). Statistical analyses were performed with Microsoft Excel 2010^®^ (Microsoft, Redmond, Washington, USA) and SSPS Statistic 24 software^©^ (IBM, Armonk, NY, USA). Pearson correlation tests were used to evaluate correlations between traits represented in graphs. Differences were defined as ‘not significant’ in cases with *p*-values > 0.1, as ‘tendentially significant’ (significant *) with *p*-values between 0.1 and 0.05, as ‘significant’ (significant **) with *p*-values between 0.05 and 0.005 and as ‘highly significant’ (significant ***) with *p*-values < 0.005. Figures were created with GraphPad Prism7^©^ (GraphPad Software, San Diego, CA, USA).

## 3. Results

### 3.1. Prolog

In a first step, we generated DC/DC_leu_ from four different leukemic cell lines and evaluated mRNA profiles under hypoxic and normoxic conditions. In the next step, we generated DC/DC_leu_ from leukemic and healthy PBMNCs as well as from WB, to simulate physiological conditions with different DC/DC_leu_-generating protocols. Furthermore, we analyzed the immune stimulating effect of these generated DC/DC_leu_ after MLC under hypoxic and normoxic conditions, evaluated the resulting anti-leukemic activity and correlated these findings with frequencies of DC/DC_leu_-subtypes.

### 3.2. AML Cell Lines’ Phenotypic and Genotypic Profiles Do Not Change under Hypoxic Culture

We compared the relative mRNA expression levels of cell line specific fusion genes as mentioned in Table 1 after five passages of AML cell lines under hypoxic and normoxic conditions. Levels of mRNA expression were analyzed by quantitative real-time RT-PCR, using glyceraldehyde-3-phosphate dehydrogenase gene (GAPDH) as the housekeeping control. Our results indicated no significant differences for the expression of these fusion genes under hypoxic vs. normoxic conditions (NB-4 ΔΔCT 3.5 vs 2.9; Mono-Mac-6 ΔΔCT 23.1 vs. 27.6; KG ΔΔCT 19.6 vs. 20.3; THP-1 ΔΔCT 101.2 vs. 114.5) (Figure 1A).

### 3.3. DC/DC_leu_ Generation from Leukemic Cell Lines with Kits Is Comparable under Hypoxic and Normoxic Conditions without Induction of Blast Proliferation 

DC/DC_leu_-generation with Kit I and Kit M was comparable in all four leukemic cell lines under hypoxic as well as normoxic conditions with significantly higher frequencies of DC/DC_leu_-subtypes found after DC/DC_leu_-generation with Kits compared to control. In general, Kit I was superior to Kit M in generating DC/DC_leu_ (including subgroups) from leukemic cell lines (Figure 1B). In the direct comparison of hypoxic and normoxic conditions, significantly more mature DCs (DC_mat_/DC^+^) could be generated in the NB-4 cell line with Kit I and Kit M under hypoxic compared to normoxic conditions (Figure 1B). Also, significantly more DC_leu_/Bla^+^ could be generated with Kit M under hypoxic compared to normoxic conditions. For the other cell lines, no significant differences could be found in the direct comparison of the amounts of generated DC/DC_leu_ as well as DC/DC_leu_ subgroups.

### 3.4. Significantly Lower Frequencies of Cells from Leukemic Cell Lines Found in S-Phase of Cell Cycles under Hypoxic Compared to Normoxic Conditions

We found significantly lower frequencies of cells in the S-phase of the cell cycle under hypoxic compared to normoxic conditions for the cell lines NB-4, KG-1 and THP-1. For the cell line Mono-mac-6, no significant differences were found. Frequencies of cells in the remaining cell cycle phases did not differ significantly (Figure 1C).

### 3.5. Generation of DC/DC_leu_ from Leukemic and Healthy PBMNCs Is Comparable under Hypoxic and Normoxic Conditions

DC/DC_leu_ were generated from healthy and leukemic PBMNCs (*n* = 9) under hypoxic and normoxic conditions with the two DC/DC_leu_-generating protocols Pici_-PGE1_ and Pici_-PGE2_ compared to control. The following patients’ blood samples were included in this analysis: Pat. 15, 16, 17, 19, 20, 21, 22, 24, 25. Further characteristics of the patients and stages of the disease are given in Table 1. Significantly higher frequencies of DC/DC_leu_ (including subgroups) were found after culture with Pici-_PGE1_ and Pici-_PGE2_ under hypoxic as well as normoxic conditions compared to control (Figure 2). Comparable results were found for healthy cell samples.

Comparable frequencies of DC/DC_leu_ could be generated with Pici-_PGE1_ and Pici-_PGE2_ from leukemic PBMNCs under hypoxic and normoxic conditions. Compositions of Pici-_PGE1_ and Pici-_PGE2_ protocols are given in Table 2. Average frequencies ± standard deviations of DC/DC_leu_ and their subtypes under hypoxic (left side) and normoxic (right side) conditions are given.

### 3.6. Generation of DC/DC_leu_ with Kits from Leukemic and Healthy WB Is Comparable under Hypoxic and Normoxic Conditions—without Induction of Blasts’ Proliferation

We defined a cut-off value of ≥ 5% DC^+^/WB as a ‘sufficient DC-generation’ from leukemic WB with Kits. Under hypoxic conditions, a sufficient DC-generation was possible in 88% of cases with Kit I, in 85% of cases with Kit M and in 18% of control cases (Figure 3A). Under normoxic conditions, an adequate and sufficient DC-generation from leukemic WB was possible in 96% of cases cultured with Kit I, in 77% of cases with Kit M and in 25% in the control group (Figure 3A). Comparisons showed no significant differences in sufficient DC-generation under hypoxic vs normoxic conditions.

A detailed analysis revealed that we generated significantly higher frequencies of DC (including subtypes) from leukemic WB with Kits compared to control—under hypoxic as well as under normoxic conditions (Figure 3B). Comparable results were found for healthy samples.

We found comparable frequencies of proliferating blasts (Bla_prol-CD71_/Bla^+^ and Bla_prol-Ipo-38_/Bla^+^) after culture of leukemic WB with Kits compared to control under hypoxic and normoxic conditions: (hypoxic conditions: %Bla_prol-CD71_/Bla^+^: control: 9.2 ± 7.3%; Kit I: 9.3 ± 10.9%, *p* < 0.2; Kit M: 12.3 ± 11.9%, *p* < 0.2; normoxic conditions: %Bla_prol-CD71_/Bla^+^: control: 9.8 ± 12.6% Kit I: 10.5 ± 11.7%, *p* < 0.2; Kit M: 10.6 ± 11.4%, *p* < 0.2).

### 3.7. Significantly Lower Frequencies of T_reg_ Cells Found after MLC^WB-DC Kits^ under Hypoxic Conditions Compared to Control

In general, we found a significantly higher activation status of immunoreactive T cell subtypes after MLC^WB-DC Kits^ compared to MLC^WB^ (e.g., T_prol-eraly_, T_non-naiv_) and reduced frequencies of T_reg_. Frequencies of T_regs_ significantly decreased after MLC^WB-DC Kit M^ and MLC^WB-DC Kit I^ vs control under hypoxic conditions (Figure 4A).

Furthermore, we found significantly higher frequencies of NK cells after MLC^WB-DC Kit M^ compared to the control, pointing to a stimulating effect of generated DC/DC_leu_ on cells of the innate immune system under hypoxic and normoxic conditions (Figure 4B).

In the direct comparison of hypoxic and normoxic conditions no significant difference were observed.

T cells enriched immunoreactive cells were stimulated with Kit I or Kit M pretreated (DC/DC_leu_ containing) WB compared to the control without pretreatment. Average frequencies of stimulated cells after MLC^WB-DC Kit I^, MLC^WB-DC Kit M^ and MLC^WB^ (control) ± standard deviation are given in Figure 4A. Frequencies of iNKT and NK cells under hypoxic and normoxic conditions are given in Figure 4B. Explanations of all analyzed immunoreactive cells subtypes are given in Table 3.

### 3.8. Kit Pre-Treated Leukemic WB, Leads to Significantly Improved Anti-Leukemic Activation after T Cell Enriched MLC, Especially in Hypoxic Conditions

We analyzed the blast lytic effect of immunoreactive cells after T cell enriched MLC^WB-DC Kits^ and MLC^WB^ under hypoxic as well as normoxic conditions. Blast lysis was evaluated after 3 h and 24 h of incubation of blast targets with effector cells after MLC.

After 24 h of incubation of target cells with effector cells, blast lysis was achieved in 72% of cases after MLC^WB-DC Kit I^, in 84% after MLC^WB-DC Kit M^ vs 38% of cases in the control group (MLC^WB^) under hypoxic conditions. Compared to the control group, significantly more cases achieved blast lysis after MLC^WB-DC Kit M^ (Figure 5A). We could see a clear advantage in the improvement of blast lysis after MLC^WB-DC Kit M^ compared to the control (Figure 5B) and in the frequency of improved lysis after 24 h although differences were not significant under hypoxic and normoxic conditions (Figure 5C).

Achieved blast lysis after MLC^WB-DC Kit I^, MLC^WB-DC Kit M^ and MLC^WB^ (control) after 24 h of co-cultures with leukemic blats (target cells). Percentage of cases with lysis after 24 h (Figure 5A), percentage of cases with improved lysis after 24 h (Figure 5B) and percentage of blasts increased/lysed after 24 h (Figure 5C) under hypoxic and normoxic conditions are presented.

### 3.9. Significant Correlation of Anti-Leukemic Reactivity and DC/DC_leu_ Subtypes under Hypoxic as Well as Normoxic Conditions

Correlating improved anti-leukemic reactivity of immune reactive cells after MLC^WB-DC Kit I^ and MLC^WB-DC Kit M^ vs MLC^WB^ (control) with frequencies of generated DC and DC_leu_ in cultures with Kits, we found significant positive correlations with generated DC^+^/WB under hypoxia and normoxia (r = 0.7; *p* < 0.01; r = 0.5; *p* < 0.02) and with DC_leu_/WB under Hypoxia (r < 0.6; *p* < 0.05), but not under normoxia with Kit M. (Figure 6). These correlations were not found for results obtained with Kit I.

## 4. Discussion

### 4.1. DC/DC_leu_-Based Immunotherapy for AML

AML is a clonal hematopoietic disorder with a high risk of relapse due to blasts’ immune escaping mechanisms, such as impaired or mistaken antigen expressions, resistance to apoptosis or other inhibitory mechanisms [33]. To address this issue, various immunotherapeutic strategies (including DC/DC_leu_-based strategies) are being explored to reactivate the antitumor immune response [34,35]. On the one hand DCs can be generated ex vivo from monocytes and loaded with different leukemia associated antigens and on the other hand DC/DC_leu_ can be generated directly from leukemic blasts presenting patients’ individual antigen repertoire [9,10,36]. Re-administration of these ex vivo generated and prepared DC/DC_leu_ have shown to increase the frequencies of leukemia-specific (T) cells in vivo and the treatment achieved stable complete remissions in elderly AML patients [37,38,39]. A new and interesting approach could be to induce the production of DC/DC_leu_ from leukemic blasts in vivo after the treatment of patients with DC/DC_leu_-inducing response modifiers [6].

### 4.2. Hypoxia, a Condition with Influence on Haematological and Immune Reactions

Hematopoietic and immunoreactive cells are exposed to changing O_2_ concentrations in the BM of about 0.1–1%, in arterial blood of about 12% and in PB of about 4–15% [20,21]. Several groups have utilized hypoxic conditions at 6% O_2_ to simulate physiological conditions ex vivo [21,40]. Up to now, no consistent effect of hypoxia on the immune system was shown [41]. Some studies suggest that hypoxia might suppress the anti-leukemic effect of immune cells, weakening the success of current anti-leukemic therapies [42]. Moreover a correlation between the hypoxic tumor microenvironment and poor response to radiation/chemotherapy in patients was seen [42]. Furthermore, physiological hypoxia could stimulate the proliferation and activation of NK cells, contributing to anti-leukemic functions [23]. Different studies have demonstrated that the effects of oxygen on immune cells depend on the tissue and the duration of hypoxia exposure. For example, while 6% O_2_ is considered as hypoxic, this oxygen concentration represents a normal physiological condition in the BM and stem cell niche [43,44]. Therefore, we used 10% O_2_ as an average of venous and arterial blood, as described previously [20,21]. Moreover, intense blast proliferation and O_2_ consumption in leukemic-PB might affect the biology of leukemic blasts [21,45] compared to standard normoxic condition. In hypoxic conditions, the expression of membrane receptors (e.g., CXCR4) and the activation of intracellular signaling cascades of pO_2_-sensitive tumor suppressor genes (e.g., MAPK, HIF1a) change [40]. Physiological hypoxia was shown to induce cell cycle arrest in the G0/G1-phase of AML blasts (cell lines and primary AML samples) by increasing the expression of the anti-apoptotic XIAP and activation of PI3K/Akt [46].

### 4.3. Generation of DC/DC_leu_ from Leukemic Cell Lines

We utilized AML cell lines in the initial phase to assess the impact of hypoxia, as previous studies have demonstrated that AML cell lines serve as valuable tools for investigating the effects of hypoxia on leukemic cell proliferation [47]. Our findings indicate that DCs can be successfully generated using DC/DC_leu_-generating protocols, irrespective of the type and the mutation status of the cell lines, under both hypoxic and normoxic conditions. Interestingly, our results revealed no significant decrease in the frequencies of generated DC/DC_leu_ and no discernible differences in the blast proliferation between the two conditions. The adaptation to low oxygen levels during DC/DC_leu_-cultures can be attributed to the activity of HIF-1α [45]. Notably, we observed higher frequencies of cells in the S-phase under normoxic conditions compared to hypoxic conditions in our experiments which might result in a lower cell proliferation activity in hypoxic compared to normoxic conditions. In summary, our study confirmed that DC/DC_leu_-generation from leukemic cell lines is feasible under hypoxic conditions, yielding comparable outcomes to those obtained under normoxic conditions.

### 4.4. DC/DC_leu_-Generation from Healthy and AML PBMNCs and WB

The generation of DC/DC_leu_ using DC/DC_leu_-generating protocols from healthy and AML PBMNCs with Pici_-PGE1_ and Pici_-PGE2_ was successful compared to controls. These data confirm previous findings described by others and us [6,9,27,30]. Interestingly, all results obtained were similar under both hypoxic and normoxic conditions. These results demonstrate that hypoxia, as a physiological condition, is not necessary for the ex vivo generation of DC/DC_leu_ for later on adoptive cell transfer.

Whole blood contains all soluble and cellular factors present in the individual AML patient, which may have activating or inhibitory influences on physiological immune reactions [19]. Therefore, we generated DC/DC_leu_ directly from healthy and leukemic WB to simulate most physiological conditions. DC/DC_leu_ generation was feasible under both hypoxic and normoxic conditions and the DC_leu_ subgroups did not significantly differ between the two conditions. Importantly, the used Kits did not induce blasts’ proliferation during cultures, thereby conforming previous findings. In the clinical context, these results support the idea that AML patients could be treated directly with Kits inducing DC/DC_leu_-generation in vivo [48,49].

### 4.5. DC/DC_leu_-Stimulation in T Cell Enriched MLC Results in Activated Cells of the Innate and Adaptive Immune System

T, iNKT, NK and CIK cells and their subsets are important mediators of innate and adaptive immune responses and their anti-tumor and anti-infections functionality are known to be activated by DC/DC_leu_ [15]. This was already confirmed previously under normoxic conditions using DC/DC_leu_ generated with Kits in T cell enriched MLC [9,14,15,30,50]. We can add that this activation was seen under hypoxic as well as normoxic conditions and resulted in comparable frequencies of different immune cell subsets. Memory T cells play a critical role in the maintenance of the complete remission of AML patients if activated in vivo [51,52]. Furthermore, cells of the innate immunity significantly increased after MLC under hypoxic and normoxic conditions.

### 4.6. DC/DC_leu_ Stimulation after T Cell Enriched MLC Results in Improved Anti-Leukemic Activity

As already shown before treatment of WB with Kits, followed by T cell enriched MLC improved anti-leukemic reactivity under normoxic conditions [25,50]. Here we show that anti-leukemic activity was improved under hypoxic vs normoxic conditions. These results might point to different killing mechanisms under hypoxic vs normoxic conditions or at least a variation of this process (e.g., slow pathway of Fas/FasL- or the fast pathway of perforin-granzyme-mediated killing). These effects might act synergistically or independently [53,54]. Additionally, the superior anti-leukemic activity observed after MLC^WB-DC Kit I^ and MLC^WB-DC Kit M^ appeared to be equally effective under normoxic conditions. However, a significantly higher number of cases with increased lysis after MLC^WB-DC Kit M^ compared to controls (MLC^WB^) was found under hypoxic versus normoxic conditions. These results suggest that hypoxia, as a physiological condition may enhance the anti-leukemic activity.

### 4.7. Correlation of Anti-Leukemic Cytotoxicity of Immunoreactive Cells Stimulated by DC/DC_leu_

We investigated the correlation between the anti-leukemic reactivity of effector cells (T cell-enriched WB treated with or without Kits) and the frequencies of various DC and immune cell subtypes. In normoxic conditions, a significant positive correlation was found between DC^+^/WB generated with Kit M and the highest anti-leukemic activity after 3 or 24 h of treatment with MLC^WB-DC Kit M^. Under hypoxic conditions, a significant correlation was observed between DC^+^/WB and DC_leu_/WB generated with Kit M and the most effective anti-leukemic reactivity of T effector cells after MLC^WB-DC Kit M^ after 3 or 24 h. Other cell subtypes did not show a direct correlation with achieved cytotoxicity. These findings support previous studies indicating a relationship between DC stimulation and the gained cytotoxicity of stimulated T cells against leukemia [55]. Moreover, the better anti-leukemic reactivity of MLC^WB-DC Kit M^ compared to MLC^WB^ could be a good explanation for the positive correlation between (DC_leu_/WB) generated with Kit M and the best anti-leukemic activity of T cells (MLC^WB-DC Kit M^) in hypoxic conditions.

## 5. Conclusions

DC/DC_leu_-generation with Kit I and Kit M was shown to increase DC/DC_leu_ frequencies and to activate different immune cells after T cell enriched MLC under hypoxic and normoxic conditions in comparable frequencies using AML cell lines, PBMNCs and WB from leukemic and healthy samples. Induced anti-leukemic reactions were shown to be superior under hypoxic vs normoxic conditions. In summary, we show that immunoreactions induced by DC/DC_leu_ might be underestimated under normoxic conditions—pointing to improved effects of DC/DC_leu_-immunotherapies in vivo in the hypoxic niches of the body. However, additional studies are needed to evaluate the impact of hypoxic conditions on the immune system and the generation of DC/DC_leu_ in vivo.

## Figures and Tables

**Figure 1 cancers-16-02383-f001:**
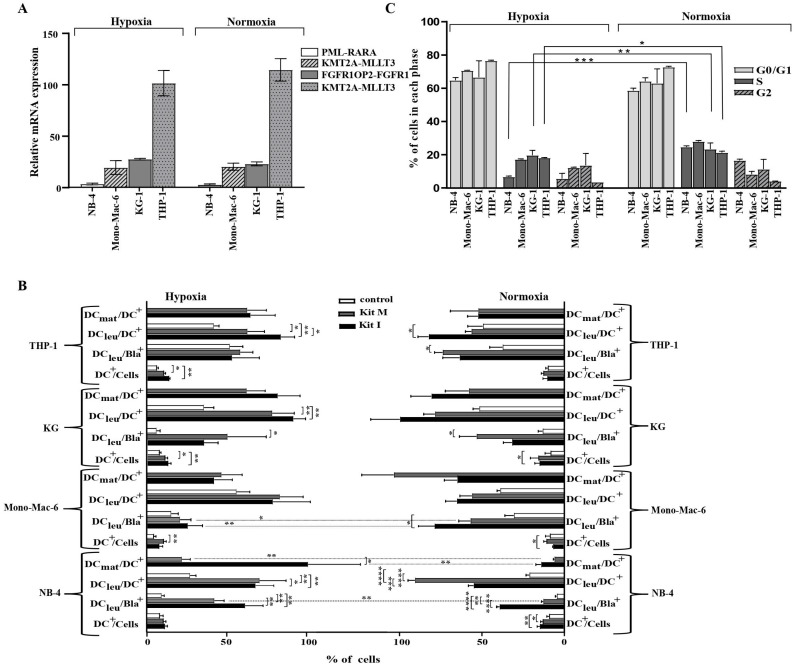
Characterization of AML cell lines and the generation of DC/DC_leu_ under hypoxic and normoxic conditions. qPCR Real Time Quantitative Polymerase chain reaction; AML acute myeloid leukemia; mRNA messenger RNA; hypoxic conditions (5% O_2_ saturation); normoxic conditions (21% O_2_ saturation); CT cycle threshold; GAPDH glyceraldehyde-3-phosphate dehydrogenase gene, DC_mat_ mature DC; DC dendritic cells; DC_leu_ dendritic cells of leukemic origin; Bla leukemic blast, hypoxic conditions (5% O_2_ saturation); normoxic conditions (21% O_2_ saturation); S synthesis phase of the cell cycles; G1 gap/growth 1 phase of the cell cycles; G2 gap/growth 2 phase of the cell cycle. * *p*-values between 0.1 and 0.05, ** *p*-values between 0.05 and 0.005, *** *p*-values < 0.005, **** *p*-values < 0.0005. (**A**)**. qPCR analyses of fusion genes** Four different AML cell lines (NB-4, Mono-Mac-6, KG-1 and THP-1) were cultured in the RPMI medium under hypoxic as well as normoxic conditions. Results show no differences between the expression of fusion genes (PML-RARA fusion gene in the NB-4 cell line, FGFR1OP2-FGFR1 fusion gene in the KG-1 cell lines, KMT2A-MLLT3 fusion gene in the Mono-Mac-6 cell line and KMT2A-MLLT3 fusion gene in the THP-1 cell line) under hypoxic (left side) or normoxic (right side) conditions. Relative expressions of fusion genes specific for each cell line were tested after five passages. The glyceraldehyde-3-phosphate dehydrogenase gene (GAPDH) was used as the housekeeping control. The y-axis shows the ΔΔCT differences between the genes of interest and the housekeeping gene. Average frequencies ± standard deviation are given. (**B**)**. DC/DC_leu_ subsets after DC/DC_leu_-generation** Results of DC/DC_leu_ subsets after DC/DC_leu_-generation from four different cell lines lines (NB-4, Mono-Mac-6, KG-1, and THP-1) with Kit I, Kit M and control (culture without added response modifiers) are given. Only in the NB-4 cell line could significantly higher frequencies of mature DCs be generated with Kit I and Kit M under hypoxic compared to normoxic conditions. For all other subsets, comparable frequencies could be generated. Average frequencies ± standard deviations are given. (**C**)**. Cell cycle analysis** Cells were fixed in ethanol, stained with propidium iodide (PI) and analysed for DNA content to determine populations in G1 and S phases of the cell cycle. Significantly lower frequencies of cells are found in the S-phase under hypoxic vs. normoxic conditions for the NB-4, KG-1 and THP-1 cell lines. Average frequencies ± standard deviation are given.

**Figure 2 cancers-16-02383-f002:**
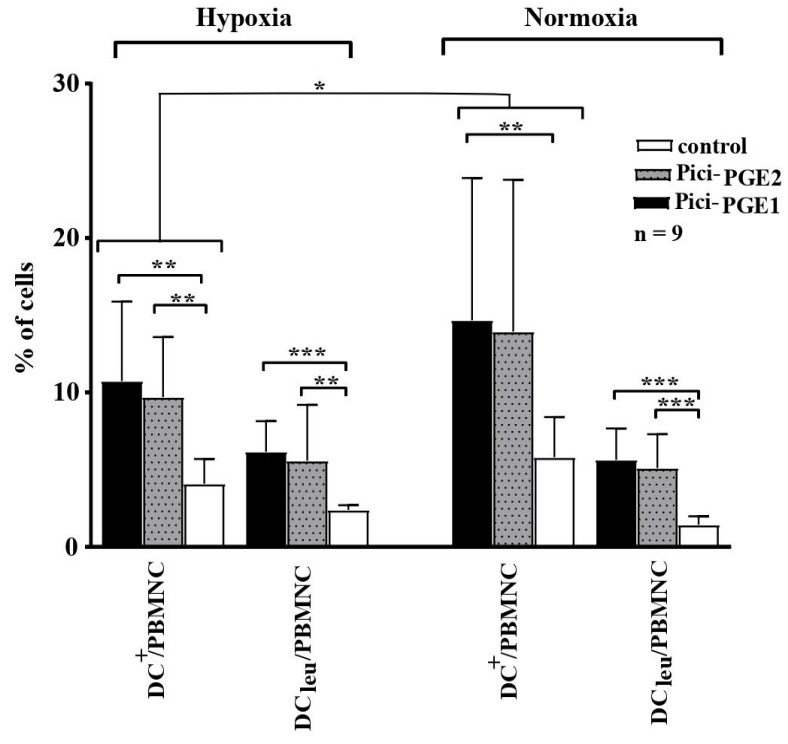
DC/DC_leu_ generation from leukemic PBMNCs under hypoxic and normoxic conditions. Hypoxic conditions (5% O_2_ saturation); normoxic conditions (21% O_2_ saturation); DC dendritic cells; DC_leu_ dendritic cells of leukemic origin; PBMNCs peripheral blood mononuclear cells. * *p*-values between 0.1 and 0.05, ** *p*-values between 0.05 and 0.005, *** *p*-values < 0.005.

**Figure 3 cancers-16-02383-f003:**
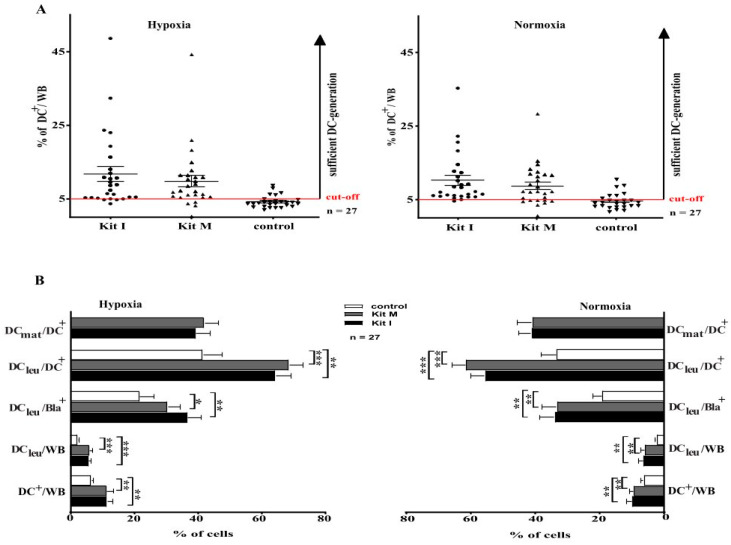
DC/DC_leu_ generation from leukemic WB under hypoxic and normoxic conditions. WB whole blood; DC dendritic cells; hypoxic conditions (10% O_2_ saturation); normoxic conditions (21% O_2_ saturation), DC dendritic cells; DC_leu_ dendritic cells of leukemic origin; DC_mat_ mature dendritic cells. (**A**). Leukemic WB samples were cultured in parallel under hypoxic and normoxic conditions with Kit I and Kit M compared to controls (without added response modifiers). Proportions of cases with ‘sufficient DC-generation’ (setting a cut-off-value at 5% DC^+^/WB) from leukemic WB were not different under hypoxic (left side) and normoxic conditions (right sight). Each dot (● ▼) characterizes DC-frequencies generated from each AML-patient in each given case. (**B**). Leukemic WB samples were cultured with Kit I, Kit M and without added response modifiers (control) under hypoxic (left side) and normoxic (right sight) conditions. Average frequencies ± standard deviation of DCs and their subtypes are given. * *p*-values between 0.1 and 0.05, ** *p*-values between 0.05 and 0.005, *** *p*-values < 0.005.

**Figure 4 cancers-16-02383-f004:**
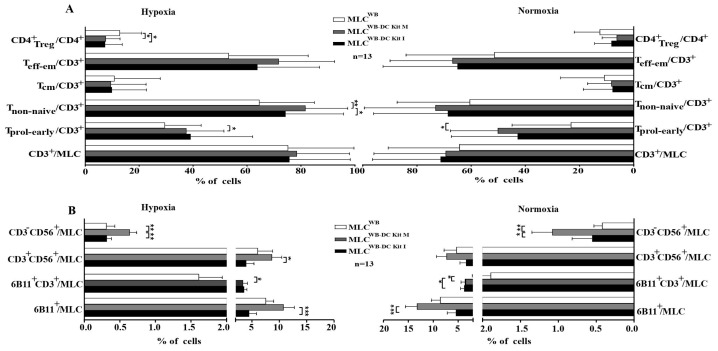
Composition of immunoreactive cells after T cell enriched MLC under hypoxic and normoxic conditions. % percentage; (**A**) shows amount of cells of the adaptive immune system, (**B**) shows amounts of cells of the innate immune system. MLC mixed lymphocyte cultures; MLC^WB-DC Kit I^ and MLC^WB-DC Kit M^ MLC with Kit I or Kit M pretreated WB, MLC^WB^ control; hypoxic conditions (10% O_2_ saturation); normoxic conditions (21% O_2_ saturation). * *p*-values between 0.1 and 0.05, ** *p*-values between 0.05 and 0.005, *** *p*-values < 0.005.

**Figure 5 cancers-16-02383-f005:**
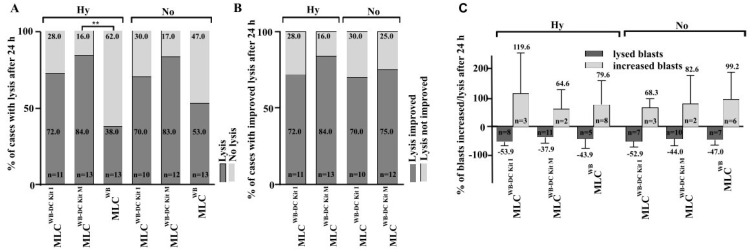
Lysis of leukemic blasts after T cell enriched MLC under hypoxic and normoxic conditions. % percentage; h hours; MLC T cell enriched mixed lymphocyte cultures; MLC^WB-DC Kit I^ and MLC^WB-DC Kit M^ MLC with Kit I or Kit M pretreated WB; MLC^WB^ control. ** *p*-values between 0.05 and 0.005.

**Figure 6 cancers-16-02383-f006:**
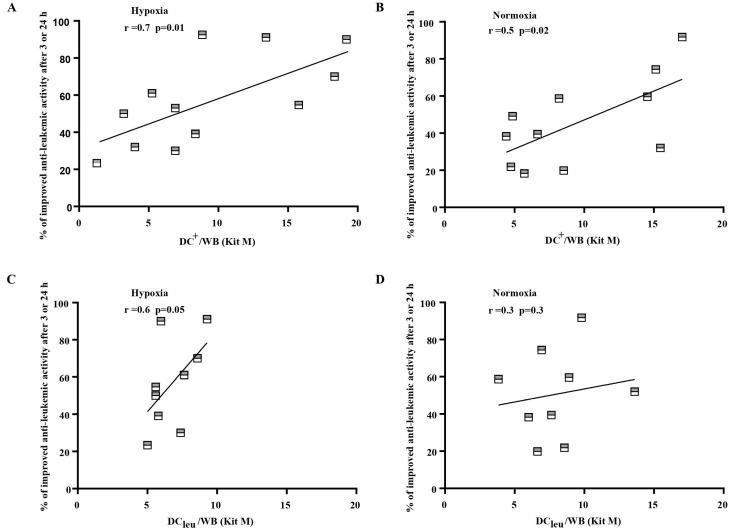
Correlation analyses of cases with improved anti-leukemic reactivity and DC and DC_leu_ under hypoxic (**A**,**C**) and normoxic conditions (**B**,**D**). DC dendritic cells; WB whole blood; DC_leu_ dendritic cells of leukemic origin. Improved anti-leukemic activity after MLC^WB-DC Kit M^ in comparison to MLC^WB^ (*y* axis) correlated positively with frequencies of generated DC^+^/WB and DC_leu_/WB with Kit M (*x* axis) under hypoxic (left) and normoxic (right) conditions. Correlation was evaluated with pearson correlation analyses.

**Table 1 cancers-16-02383-t001:** Characteristics of AML Patients and AML Cell Lines.

Pat.#	Age at dgn.	Sex	Subtype AML FAB	Blast Phenotype (CD)	Blasts (%) in PB *	ELN-Risk Stratification
**Patients with First Diagnosis**						
P1	61	f	p/M5	13, 33, **34**, 64, **117**	40	adverse
P2	52	m	p/M2	13, 33, **117**	96	favorable
P3	79	m	p/M5	13, 33, **34**, **117**	70	favorable
P5	59	f	s/M4	33, 13, 14, 65, **117**, **34**	15	favorable
P6	60	m	s/M4	33, 13, 14, 65, **117**, **34**	81	favorable
P7	73	f	p/M4	**117**, 33, 61	14	intermediate
P8	64	m	p/ M1	13, 33, **34**, **117**	28	favorable
P9	36	f	p/M1	33, 65, 15, **34**, **117**	63	favorable
P10	21	m	p/M5	**34**, **117**, 33, 13	33	adverse
P11	44	m	p/M4	**34**, **117**, 33, 13	50	intermediate
P12	54	m	p/M4	33, 64, 11, 4, **56**	7	favorable
P13	78	f	p/M2	**34**, 33, 15, **65**	61	adverse
P14	78	f	s/n.d.	**34**, 64, 14, 33, 13	30	adverse
P16	39	m	p/M2	**15**, **117**	30	favorable
P17	33	f	p/M2	33, 13, **34**, **117,** 56	83	favorable
P18	73	m	p/M2	33, 13, **34, 117**	84	adverse
P21	69	m	s/n.d.	13, 33, **117**	38	adverse
P22	66	m	n.d.	**34**, 13, 33, **117**	12	intermediate
P23	63	f	p/M4	**117**, **13**, 64	12	intermediate
P24	75	m	p/M5	**117,** 33, 64	40	adverse
P25	77	m	p/M5	13, **34**, 33, 64	55	n.d.
P26	44	m	p/M4	**34**, **117,** 13, 33, 64	46	n.d.
P27	77	m	p/M1	13, 33, **34**, **117**	50	intermediate
P28	50	f	s/M2	13, 33, **34**, **117**	43	n.d.
P29	55	f	p/M5	13, **34**, 7, **117**	87	adverse
P30	74	m	n.d.	13, 33, **117**, **34**	53	n.d.
P31	56	m	p/M4	13, 33, **34**, **117**	66	intermediate
P33	47	f	s/n.d.	**117**, **33**, 56, 4	61	favorable
P34	63	m	s/M5	4, 56, **14**, **34**	25	Favorable
**Patients with relapse**						
P4	59	f	p/M5	33, **34**, **117**, 13, 64, 7, 56	58	
P19	70	m	s/M1	13, 33, **117**, **34**, 56, 64, 14	80	
P20	78	m	p/M1	13, 33, **34**, **117**	60	
P32	47	m	p/M2	13, **34**, **117**	56	
**Patients with relapse after hematopoietic stem cell transplantation**						
P15	60	f	p/M1	13, 33, **34**, 56, 14	30	
**AML Cell Line**	**Subtype AML (FAB)**		**Blast phenotype (CD)**	**Fusion gene**	**Fusion gene Original Source**	
NB-4	M3		13, **15**, **33**	PML-RARA	PB	
Mono-Mac-6	M5		**13**, 14, 15, 33, 68	KMT2A-MLLT3	PB	
KG-1	M4		13, **15**, **33**	FGFR1OP2-FGFR1	PB	
THP-1	M5		**33**, 13, 14, 15	KMT2A-MLLT3	PB	

Pat. # Patient’s number; f female; m male; CD cluster of differentiation; s secondary AML; p primary AML; bold blasts markers used to detect DC_leu_; n.d. no data; * evaluated by flow cytometry; PB peripheral blood.

## Data Availability

The data published in this study is openly available in a public repository with a permanent identifier, such as a DOI.

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
