# Peer review of "Anti-Leukemic Effects Induced by Dendritic Cells of Leukemic Origin from Leukemic Blood Samples Are Comparable under Hypoxic vs. Normoxic Conditions"

_cancers, 2024, doi:10.3390/cancers16132383_

Round 1

Reviewer 1 Report

Comments and Suggestions for Authors

This article investigates the effect of hypoxia on DC generation and tumor cell-lysis efficacy compared to normoxic conditions.

Additionally, the authors compared the efficacy of various culture conditions, including immunomodulatory kits, for generating leukemia-derived DCs (DCleu).

They concluded that the activation of immune cells and anti-leukemic activities improved when T cell-enriched mixed lymphocytes were co-cultured with DC/DC leu generated with Kit I and Kit M. Additionally, the anti-tumor activity was better with DC/DC leu in hypoxic conditions, suggesting the presumptive role of increased iNKT cells.

In Figure 4, please directly compare each immune cell between hypoxic and normoxic conditions. The authors suggested that iNKT cells increase in the hypoxic condition compared with the normoxic condition; please provide the statistical analyses between the two conditions. If the authors provide a summarized table, it would be helpful to understand the differences between the two conditions.

In Figure 5, what factors contribute to determining the blast response? What are the differences in the immune cells between MLC WB-DC Kit I and Kit M? In Figure 4, cells with Kit M had similar CD4+ T regs but significantly more T eff-em, NK cells, NKT cells, and iNKT cells than those with Kit I. This is not consistent with the findings in Figure 5B, which showed better blast lysis activity with the cells cultured in Kit I.

Is the increase in the percentage of cases with lysis after 24 hours (Figure 5A) due to the low tumor cell lysis in the MLC WB under hypoxic conditions? It is hard to conclude that the hypoxic condition enhanced the tumor cell lysis by MLC-WB-DC Kit I or Kit M.

Additionally, the figure legends are too small to read, and the figures are difficult to differentiate because of their similar color and pattern. It would be great if the figures could be made more easily recognizable at a glance.

Comments on the Quality of English Language

English language is fine, but it may need minor correction.

Author Response

Dear reviewer,

first, we want to thank you for reviewing our manuscript. Please find below our response to all the points of criticism (given in italics). We thank you in advance for all the work on your side. Please do not hesitate to contact us if questions arise.

We hope that our manuscript will be accepted for publication in your Journal and look forward to your soon reply.

Best regards,
Dr. Daniel Amberger und Prof. Helga Schmetzer

Reviewer 2 Report

Comments and Suggestions for Authors

Interest to the readership

This paper would interest people investigating DC immune therapies for treating AML or the hypoxic nature of the AML microenvironment.

Novelty

There are a few papers looking at in vitro generation of DC in hypoxic conditions and of these, fewer look at generation of DCs from AML samples

Significance to the field

The paper has low significance to the field as it is incremental work describing in brief the in vitro generation of DC from AML patients. It is not clear how translating these cells in to a clinical pathway would be superior to other trials.

Major comments

The title does not reflect the conclusions from the paper.

Throughout the paper the results need to be presented more clearly.

L84; Positive or negative selection for isolating CD3 cells.

Examples of the flow cytometry gating on the populations of derived DC from cell lines would help readers understand what a DC derived from a myeloid derived cell line looks like. What is the actual cell surface phenotype of these AML cell line derived DC? Are they more or less activated cells rather than DC? What is the DC criteria?

L110-124; Showing flow plots for the patient derived DC is necessary as the % of cells described in the text do not add to 100%. What are the remaining ungated cells in the denominator? What do these DC populations look like?

L204; Are the CD3 cells autologous or allogeneic? This is important to understand how these studies will reflect how these cells may be used in the clinic in the future.

L211; What O2 conditions are the MLRs incubated in?

Table 3; What does the column abbreviation refer to? Is it what is described in line 200? The panel of antibodies and gating used to identify the subgroups needs to be included. In the DC world subgroups may be thought to refer to the DC1-DC6 nomenclature and from this table it appears that something else is meant. This needs to be clearly articulated.

L221;Could this description also be outlined in an equation to make it clearer?

L243; were any of the stats analysed as paired samples when normoxic and hypoxic from the same patient were compared.

L268; the methods describe b-actin as the housekeeping gene but results suggest GAPDH?

L270; What do the error bars represent? How many samples and repeats?

L279; Can the response of Monomac to the treatment with KitI be explained as the figure 1C indicates that there is a significant difference between O2 conditions that is not described in the text.

Section 3.3 and 3.4; Table 3 does not described the phenotype of the DC population differentiated from the AML lines. This needs to be made more explicit.

What is the viability of the cells after the different treatment and those being used in the MLC?

Figure 2; How many different AMLs and healthy cells? Where the AMLs all from the same subtype (If n=9 then this is not providing an explanation of whether all M5s or prognostic subtype etc were used?? The control conditions need better/clearer explanation?

L358 No unfilled triangle on the graph

Figure 5; What does the % of cases mean? Or is this % of lysis? Could lysis and proliferation be distinguished? Were there apoptotic cells present?

Figure 6; % of cases again seems unclear in this context.

What were the target cells in the anti-leukemic activity studies – autologous or allogeneic or a cell line?

Did the hypoxic conditions get confirmed by testing expression of HIF1a? Would this be a good control?

The data is not presented in a way that other researchers could repeat the studies as there are too many gaps in the information (particularly with respect to what is a DC as defined by the authors)

Minor comments

L56: I am unsure how one ‘downregulates frequency’ – perhaps the meaning is ‘the frequency is reduced’

L65-68; These sentences seem to say the same thing and do not add anything substantial to the narrative.

L70; What does ‘mods’ mean

L429 Pearson rather than Person

Summary of paper

Overall the conclusion that DC populations can be generated from AML samples is not new. The appeared to be no real impact of hypoxia compared to normoxia.

Comments on the Quality of English Language

Excellent quality of English with minor edits/corrections required. Some are noted above.

Author Response

.Dear reviewer,

first, we want to thank you for reviewing our manuscript. Please find below our response to all the points of criticism (given in italics). We thank you in advance for all the work on your side. Please do not hesitate to contact us if questions arise.

We hope that our manuscript will be accepted for publication in your Journal and look forward to your soon reply.

Best regards,
Dr. Daniel Amberger und Prof. Helga Schmetzer
